# Pirating conserved phage mechanisms promotes promiscuous staphylococcal pathogenicity island transfer

**Janine Bowring[1†], Maan M Neamah[1,2†], Jorge Donderis[3†], Ignacio Mir-Sanchis[4‡], Christian Alite[3], J Rafael Ciges-Tomas[3], Elisa Maiques[3,4], Iltyar Medmedov[3], Alberto Marina[3]*, José R Penadés[1]***

[1]Institute of Infection, Immunity and Inflammation, College of Medical, Veterinary and Life Sciences, University of Glasgow, Glasgow, United Kingdom; [2]Department of Microbiology, Faculty of Veterinary Medicine, University of Kufa, Kufa, Iraq; [3]Instituto de Biomedicina de Valencia (IBV-CSIC) and CIBER de Enfermedades Raras, Valencia, Spain; [4]Departamento de Ciencias Biomédicas, Universidad CEU Cardenal Herrera, Valencia, Spain

**Abstract** Targeting conserved and essential processes is a successful strategy to combat enemies. Remarkably, the clinically important *Staphylococcus aureus* pathogenicity islands (SaPIs) use this tactic to spread in nature. SaPIs reside passively in the host chromosome, under the control of the SaPI-encoded master repressor, Stl. It has been assumed that SaPI de-repression is effected by specific phage proteins that bind to Stl, initiating the SaPI cycle. Different SaPIs encode different Stl repressors, so each targets a specific phage protein for its de-repression. Broadening this narrow vision, we report here that SaPIs ensure their promiscuous transfer by targeting conserved phage mechanisms. This is accomplished because the SaPI Stl repressors have acquired different domains to interact with unrelated proteins, encoded by different phages, but in all cases performing the same conserved function. This elegant strategy allows intra- and inter-generic SaPI transfer, highlighting these elements as one of nature's most fascinating subcellular parasites.
DOI: https://doi.org/10.7554/eLife.26487.001

**\*For correspondence:** amarina@ibv.csic.es (AM); joser.penades@glasgow.ac.uk (JéRPé)

[†]These authors contributed equally to this work

**Present address:** [‡]Department of Biochemistry and Molecular Biology, The University of Chicago, Chicago, United States

## Introduction

The *Staphylococcus aureus* pathogenicity islands (SaPIs) are the prototypical members of an extremely common and recently identified family of mobile genetic elements, the phage-inducible chromosomal islands (PICIs) (*Martínez-Rubio et al., 2017*; *Penadés and Christie, 2015*). The SaPIs are clinically relevant because they carry and disseminate superantigen genes, especially those for toxic shock toxin and enterotoxin B. They are very widespread among the staphylococci and are exclusively responsible for menstrual toxic shock, a rare but important human disease. In the absence of a helper phage they reside passively in the host chromosome, under the control of a global SaPI-coded repressor, Stl, a DNA-binding protein whose sequence is rather poorly conserved among the different members of the SaPI family (*Tormo-Más et al., 2010*).

Following infection by a helper phage or induction of a helper prophage, they excise, replicate extensively, and are packaged in phage-like particles composed of phage virion proteins, leading to very high frequencies of inter- as well as intrageneric transfer (*Novick et al., 2010*; *Penadés and Christie, 2015*). In previous work we demonstrated that SaPI de-repression is effected by specific phage proteins that bind to Stl, disrupting the Stl-DNA complex and thereby initiating the excision-replication-packaging (ERP) cycle of the islands (*Tormo-Más et al., 2010*). Different SaPIs encode different Stl repressors, so each SaPI targets a different phage protein for its de-repression. Thus,

**eLife digest** Many harmful microbes can produce different molecules that make them more effective in causing and spreading diseases. These molecules can also be obtained from 'mobile genetic elements' that can be transferred between bacteria within a population. Pathogenicity islands are one such type of mobile genetic element and are very common among bacteria known as staphylococci. They spread toxin-encoding genes between bacteria, including one that can lead to a condition called toxic shock syndrome in humans.

Pathogenicity islands are normally found within the DNA of the bacteria, where they are deactivated by specific repressor proteins. However, in the presence of another type of mobile genetic element – the bacteriophages – the repressor proteins start to interact with specific proteins encoded by the bacteriophages. This allows the pathogenicity islands to become active and spread to other bacteria.

Previous research has shown that in the bacterium known as *Staphylococcus aureus*, different pathogenicity islands have different repressors. Scientists therefore assumed that the repressors are only able to interact with certain bacteriophage proteins. However, since pathogenicity islands are widespread in nature, it could be possible that they use other ways to hijack the bacteriophage machinery to ensure their transfer.

To test this hypothesis, Bowring et al. studied two types of pathogenicity islands in *S. aureus* and revealed that their two different repressors did not interact with specific bacteriophage proteins as previously hypothesized. Instead, each repressor could interact with multiple bacteriophage proteins that had a variety of different structures, including proteins from completely different bacteriophages.

Bowring et al. also discovered that each of the analyzed repressor proteins did not actually recognize any specific shared structural features on the bacteriophage proteins, but rather evolved to target proteins that play the same role in various bacteriophages. This suggests the repressors target a specific process rather than a single protein. This strategy allows them to be transferred within the same species, but also between different ones.

A next step will be to better understand how a repressor can recognize structurally unrelated proteins, and establish what evolutionary forces are driving this phenomenon. A deeper knowledge of how pathogenicity islands spread between staphylococci is vital to understand how these bacteria can become resistant to treatments such as antibiotics.

DOI: https://doi.org/10.7554/eLife.26487.002

the inducers for SaPIbov1, SaPIbov2 and SaPI1 correspond to the phage trimeric dUTPase (Dut), 80α ORF15 and Sri, respectively (*Tormo-Más et al., 2010, 2013*). Since SaPIs require phage proteins to be packaged, this strategy couples the SaPI and phage cycles, but imposes a significant transmission cost on the helper phages (*Frígols et al., 2015*). Importantly, although phages carrying mutations in the genes encoding the aforementioned SaPI inducers can be propagated in the lab, these mutations have a fitness cost when the mutant phages compete with the wild-type phages in the same conditions (*Frígols et al., 2015*), which indicates that the phage coded SaPI inducers provide an important function for the phages in nature.

We recently proposed that phages could easily overcome this SaPI imposed cost using two complementary strategies that result in phages with reduced or null capacity to induce the islands (*Frígols et al., 2015*). On the one hand, phages can encode allelic variants of the SaPI inducers with reduced affinity for the SaPI coded Stl repressor. On the other hand, some phages seem to overcome SaPI induction by replacing the phage-encoded SaPI inducing gene by another one encoding an analogous protein (an unrelated protein that performs the same biological function). Although experiments performed in the laboratory suggest that in response to these strategies SaPIs can antagonistically coevolve by inactivating their Stl repressors, this strategy superimposes a high cost for the bacteria, associated with an uncontrolled SaPI replication (*Frígols et al., 2015*), so it is unlikely that this occurs in nature.

A recent study, however, questioned the idea that phages could overcome the SaPI tyranny by replacing the SaPI inducing gene by another one encoding a functionally related protein. While all

the staphylococcal *S. aureus* phages encode Duts; some encode dimeric and others trimeric Duts, never both (*Frígols et al., 2015*). Importantly, dimeric and trimeric Duts are completely unrelated both in sequence and structure, representing a nice example of convergent evolution (*Penadés et al., 2013*). While the 80α and ϕ11 phage-encoded trimeric Duts were initially described as the SaPIbov1 inducers (*Tormo-Más et al., 2010*; *2013*), the dimeric Dut from phage ϕNM1 also induces SaPIbov1 (*Hill and Dokland, 2016*; *Hill et al., 2017*). The fact that both dimeric and trimeric Duts induce SaPIbov1 raised the interesting possibility that the Stl repressors could target different phage proteins, significantly increasing the capacity of the SaPIs to be induced and transferred. This result also raised other interesting questions about the SaPIs: is this phenomenon exclusive of SaPIbov1 or are other SaPIs also induced by unrelated proteins? If that was the case for specific SaPIs, are these unrelated proteins always performing the same function for the phages or conversely it is possible that a specific SaPI repressor interacts with proteins performing unrelated functions? And finally, what is the molecular mechanism by which the SaPI-encoded Stl repressors interact with different proteins?

Here we set out to answer all these questions and have demonstrated that it is more complicated than expected for the phages to overcome the SaPIs superimposed tyranny. Our results provide evidence of inter-species PICI transfer in nature. We have also deciphered the molecular mechanism used by the SaPIs to hijack the helper phage machinery in order to get high intra- and inter-generic transference: instead of interacting with specific partners, SaPIs have evolved a fascinating strategy that promotes their high transfer by pirating conserved phage mechanisms.

## Results

### The SaPIbov1 Stl repressor interacts with the ϕO11 dimeric Dut protein

What is the mechanism by which the SaPIbov1 repressor interacts with apparently unrelated proteins? Obviously, and since the trimeric and dimeric Duts perform the same biological function, the most likely scenario would be the existence of a conserved domain in the phage-encoded proteins that would be recognised by the SaPIbov1 coded Stl repressor. The structure of the phage 80α and ϕ11 coded Duts has recently been solved (*Leveles et al., 2013*; *Tormo-Más et al., 2013*). Moreover, in-depth structural, genetic and biochemical studies have demonstrated that the trimeric Dut domains IV, V and VI are involved in SaPIbov1 Stl recognition (*Maiques et al., 2016*; *Tormo-Más et al., 2010*, *2013*). To know whether similar domains are present in the dimeric Duts, we initially addressed the following question: does the SaPIbov1 Stl interact just with the ϕNM1 dimeric Dut or can it interact with other phage coded dimeric Duts? To solve this question, we analysed the ϕO11 dimeric Dut. As occurred with the trimeric Duts (*Tormo-Más et al., 2010*), the dimeric ϕNM1 and ϕO11 Duts are basically identical except in a divergent central region (*Figure 1—figure supplement 1*). Interestingly, the ϕO11 dimeric Dut also induces the SaPIbov1 and SaPIbov5 cycles (*Figure 1A*). Note that SaPIbov5 was also included in these studies because it encodes the same Stl repressor as SaPIbov1, with both islands being induced by the same helper phages (*Carpena et al., 2016*; *Viana et al., 2010*). Expression of the ϕO11 dimeric Dut (from the P*cad* promoter in expression vector pCN51) in a SaPIbov1 or SaPIbov5 positive strain demonstrated that this protein is sufficient to induce the SaPI cycles. Thus, when overexpressed, the cloned ϕO11 dimeric *dut* induced SaPIbov1 and SaPIbov5 excision and replication (*Figure 1A*).

In all 3 characterised SaPIs (SaPI1, SaPIbov1 and SaPIbov2), Stl blocks SaPI induction by binding to the SaPI *stl-str* divergent region, blocking transcription of most of the SaPI genes. SaPI de-repression occurs after a direct protein-protein interaction between the cognate phage inducer and the SaPI coded Stl repressor (*Tormo-Más et al., 2010*). To test if the mechanism involving the ϕO11 dimeric Dut in SaPIbov1 induction matches with that previously reported for the other SaPIs, we first demonstrated that ϕO11 Dut induces *xis* expression, which normally is repressed by Stl. This was confirmed using plasmid pJP674, which carries a β-lactamase reporter gene fused to *xis*, downstream of *str* and the Stl$_{SaPIbov1}$-repressed *str* promoter, and also encodes Stl$_{SaPIbov1}$ (see *Figure 1B*). The cloned ϕO11 *dut* gene was introduced on vector pCN51 and expression was tested in the presence or absence of an inducing concentration of CdCl$_2$. Induction of ϕO11 *dut* strongly increased β-lactamase expression from the *str* promoter (*Figure 1B*). Moreover, the predicted protein–protein interaction between the ϕO11 Dut and the Stl$_{SaPIbov1}$ repressor was confirmed by co-expression and

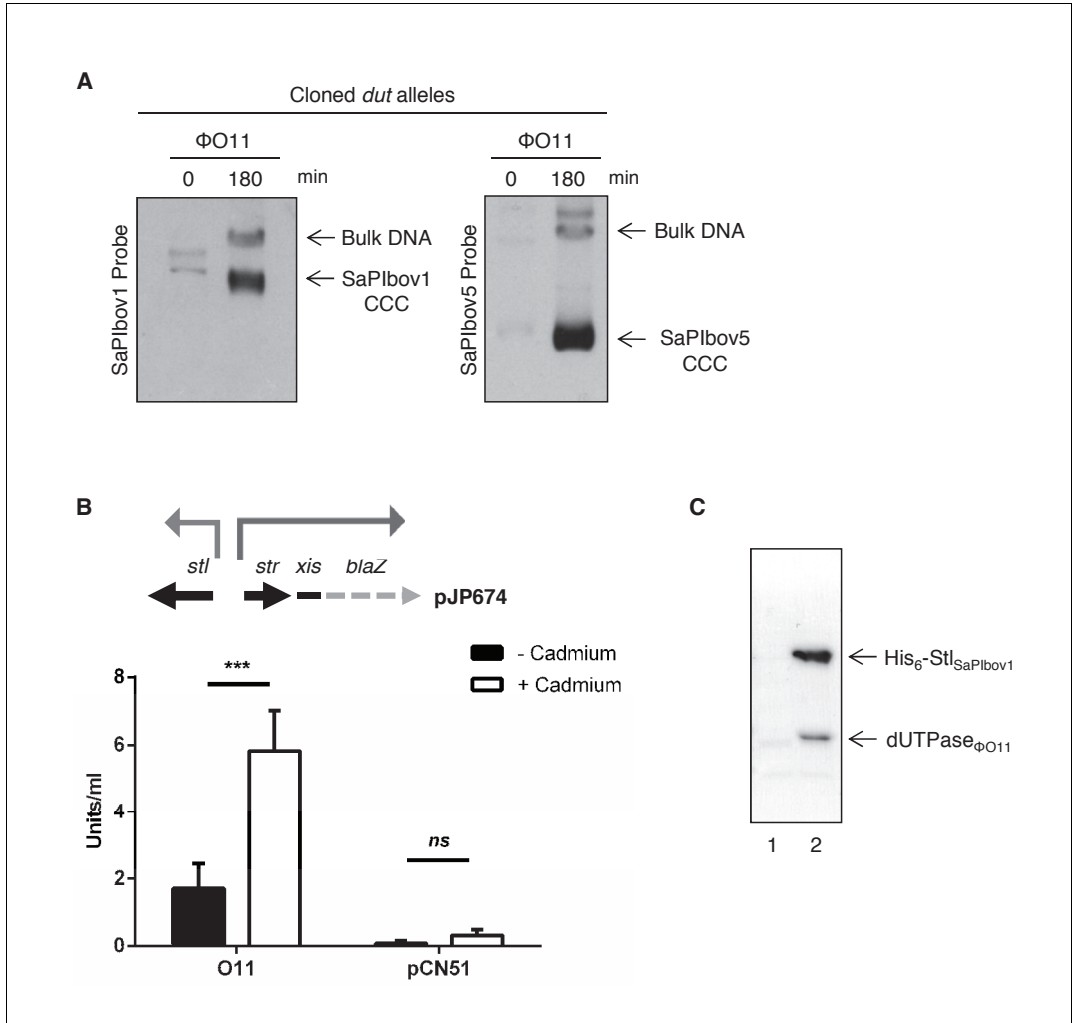

**Figure 1.** Induction of SaPIbov1 and SaPIbov5 by the ɸO11 dimeric Dut. (**A**) SaPIbov1 and SaPIbov5 excision and replication following induction of the cloned ɸO11 *dut* gene. Strains JP6774 and JP11634, containing SaPIbov1 and SaPIbov5 respectively, were complemented with a plasmid expressing the 3xFLAG-tagged ɸO11 dimeric Dut. Samples were isolated at 0' or 3 hr after induction with 0.5 µM CdCl$_2$ and Southern blots were performed using a probe for the SaPIbov1/SaPIbov5 integrase. The upper band is 'bulk' DNA, including chromosomal, phage, and replicating SaPI. CCC indicates covalently closed circular SaPI DNA. In these experiments, as no helper phage was present, the excised and replicating SaPI DNA appears as part of the bulk DNA or as CCC molecules, rather than the linear monomers that are seen following helper phage-mediated induction and packaging. (**B**) Derepression of *str* transcription by ɸO11 Dut expression. The diagram represents a schematic of a *blaZ* transcriptional fusion generated in pJP674. β-lactamase assays were performed on strains containing pJP674 together with a pCN51-derived plasmid expressing the ɸO11 Dut (JP14818) or the empty pCN51 control (JP15105). Samples were taken after 5 hr in the absence or following induction with 5 µM Cadmium. All data is the result of five independent experiments. Error bars represent SEM. A 2-way ANOVA with Sidak's multiple comparisons test was performed to compare mean differences within rows. Adjusted *p* values were as follows: ɸO11 = 0.0004[***], pCN51 = 0.9579[ns]. *ns*, not significant. (**C**) Affinity chromatography of the ɸO11 Dut for the His-tagged SaPIbov1 Stl. Strains were induced with 10 mM isopropyl-β-d-thiogalactoside (IPTG) and samples taken at 3 hr. Cells were disrupted and expressing proteins were applied to a Ni$^{2+}$ column and eluted. Lane 2, elution fraction for His$_6$-Stl$_{SaPIbov1}$ and Dut$_{ɸO11}$ (JP14832). Lane 1, corresponding elution fraction for Stl$_{SaPIbov1}$ and Dut$_{ɸO11}$ (JP14833, no His$_6$-tag). Proteins were confirmed by Mass Spectrometry analysis.

DOI: https://doi.org/10.7554/eLife.26487.003

The following source data and figure supplement are available for figure 1:

**Source data 1.** β-lactamase assay data and statistical analysis for the dimeric ɸO11 Dut.
DOI: https://doi.org/10.7554/eLife.26487.005

*Figure 1 continued*

**Figure supplement 1.** Sequence alignment for the φNM1 (accession number ABF73061) and φO11 (accession number EGA96175) dimeric dUTPases.
DOI: https://doi.org/10.7554/eLife.26487.004

affinity purification of His$_6$-Stl$_{SaPIbov1}$ and untagged φO11 Dut proteins. It was possible to co-purify a complex between His$_6$-Stl$_{SaPIbov1}$ and φO11 Dut (*Figure 1C*). The identity of each of these bands was confirmed by amino acid sequencing and mass spectrometry. We conclude from these results that dimeric φO11 Dut induces the SaPI cycle using the same mechanism described for the unrelated trimeric Dut proteins. Moreover, and although this is not the scope of this study, these results also involve the dimeric Duts in SaPI signalling.

## The SaPIbov1 Stl repressor has different interacting domains

It is predicted that dimeric and trimeric Duts acquire a completely unrelated fold (*Penadés et al., 2013*). However, since both dimeric and trimeric Duts perform the same enzymatic activity, we hypothesised that these proteins could have conserved domains responsible for the interaction with Stl. To test this hypothesis, and since the structure of the staphylococcal phage-encoded dimeric Duts remains unsolved, the structure of the dimeric φO11 Dut in complex with the nonhydrolyzable dUTP analog α,β-imido-dUTP (dUPNPP) and Mg$^{2+}$ was determined at 2.1 Å resolution (*Supplementary file 1*). The crystal structure showed 2 molecules in the asymmetric unit organized as a homodimer (*Figure 2*). The structure shows that φO11 is an all-helix protein composed of only seven α-helices (α1, residues 7–23; α2, 29–47; α3 61–82; α4, 86–98; α5, 104–108; α6, 110–121 and α7, 127–141) per protomer. Interestingly, the φO11 dimeric Dut has a 'compact' conformation compared to counterparts in other organisms (which encompass ten or more α-helices) (*Figure 2* and *Figure 2—figure supplement 1*). Nevertheless, the φO11 protomer presents the characteristic structural core of dimeric Duts composed of four helices (α1-α3 and α7 in φO11) that conforms the active centre where the nucleotide binds (*Figure 2*). In the φO11 dimer both active centres are oriented towards the same molecule face, forming a long channel that accommodates two molecules of dUPNPP. The rest of the protomer is placed on the opposite molecule face (residues 83–138), which corresponds to the divergent region in phage-encoded dimeric Duts but also adopts a helical fold (helices α4-α6 in φO11) (*Figure 2* and *Figure 2—figure supplement 1*).

Since in the trimeric Duts the motifs IV, V and VI are essential for interaction with the Stl repressor (*Maiques et al., 2016*; *Tormo-Más et al., 2013*), we looked for the presence of structural elements with similar topology in the dimeric Dut. As could be anticipated by the difference in folding between the trimeric (all-beta) and dimeric (all-alpha) proteins, none of these motifs are present in the φO11 Dut (*Figures 2* and *3*). In the trimeric Duts, these three motifs place together surrounding the nucleotide in the active centre (*Figure 3*), thus we wondered whether the Stl recognition site was generated spatially by the disposition of specific residues provided by these three motifs rather than by the motifs themselves. To check this possibility we spatially compared the active sites of both types of Duts by superimposing the nucleotide-binding sites of the trimeric 80α and dimeric φO11 phagic Duts (*Figure 3*). As was previously observed in the comparison of the active centres from other dimeric and trimeric Duts (*Harkiolaki et al., 2004*), the way of dUTP recognition and binding is completely different in both Dut types, not only in the orientation of the plane of the uracil moiety, which showed a relative rotation of more than 75°, but also in the disposition of the phosphates. In trimeric Duts, the α-phosphates acquire a *gauche* catalytic-competent geometry (*Kovári et al., 2008*) meanwhile a *trans* conformation is observed in the dimeric φO11 Dut (*Figure 3*). Furthermore, the β and γ phosphates differ in their relative disposition, chelating a single divalent metal in the trimerics, versus two in the dimerics (*Hemsworth et al., 2013*) (*Figures 2* and *3*). Therefore, the active centres in both types of enzymes show divergent architecture and, consequently, the spatial disposition of the residues surrounding the nucleotides, including those provided by motif IV, V and VI, is completely different. Taken together, these results strongly suggest that the SaPIbov1 Stl repressor has different interacting domains/ways to recognise the unrelated trimeric and dimeric Duts.

To go further with these analyses, we generated a set of deletional mutants in the SaPIbov1 Stl repressor, with the idea that some of these mutants would specifically affect the interaction of the

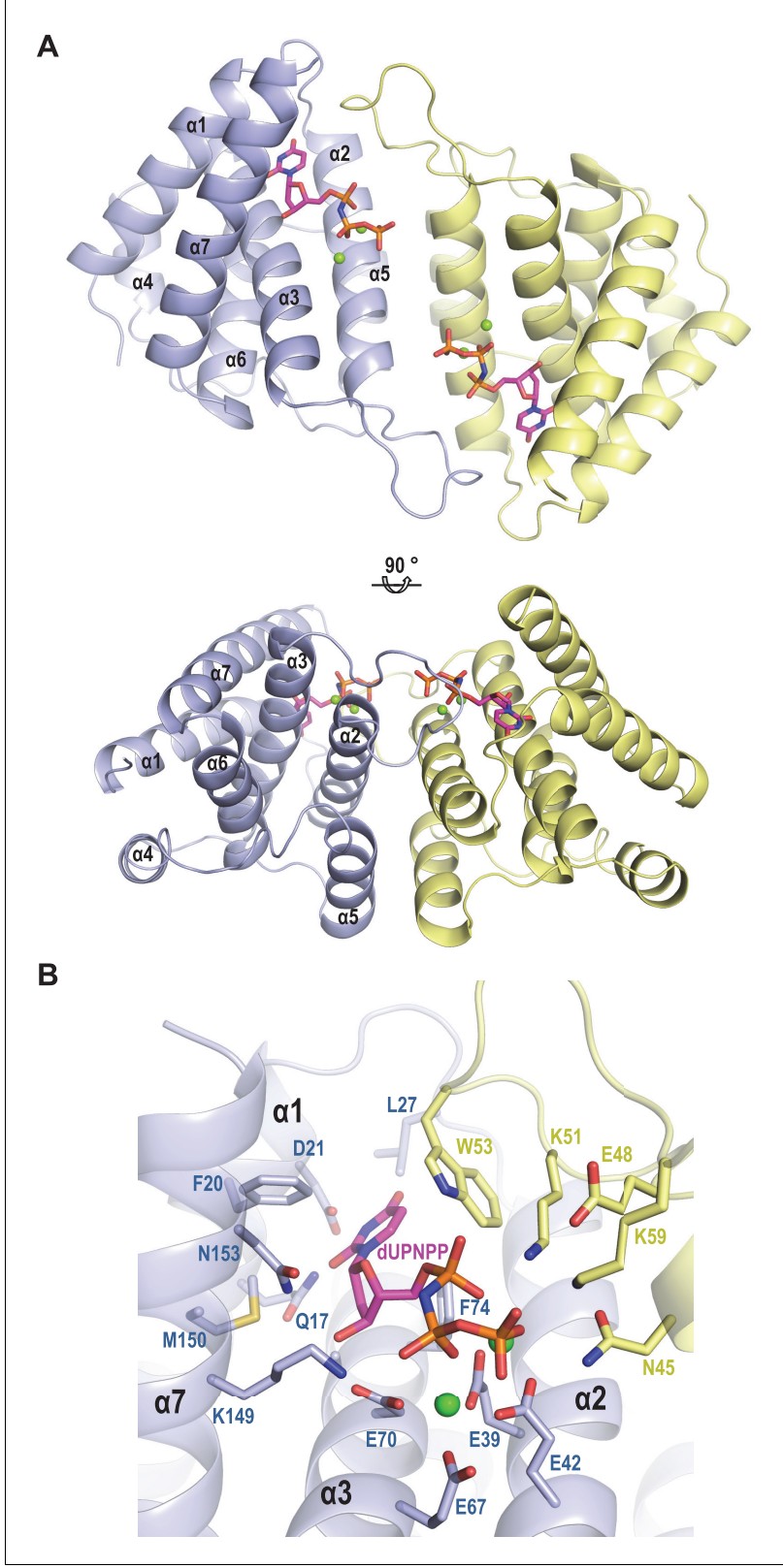

**Figure 2.** Structure of φO11 dimeric Dut. (**A**) Cartoon representation the φO11 Dut dimer (protomer in blue and yellow). The secondary structural elements are numbered. A molecule of dUMPPNP and two Mg ions represented in stick and sphere, respectively, occupy the active centre of each protomer. Two orthogonal views of the dimer are shown. (**B**) Close view of φO11 Dut active centre. The substrate dUPNPP is represented in stick with carbon

*Figure 2 continued on next page*

*Figure 2 continued*
atoms in magenta. The residues interacting with the substrate are labelled and represented in stick with carbon atoms coloured according to the protomer to which they correspond. Nitrogen, oxygen, phosphorus atoms are coloured in dark blue, red and orange, respectively. The Mg ions are represented as green spheres.
DOI: https://doi.org/10.7554/eLife.26487.006
The following figure supplement is available for figure 2:

**Figure supplement 1.** φO11 Dut represents a reduced version of dimeric Duts.
DOI: https://doi.org/10.7554/eLife.26487.007

SaPIbov1 Stl repressor with one of the Dut types under study, but not with the other. Sequence analysis and in silico modelling indicates that SaPIbov1 Stl is mainly an α-helical protein composed of a N-terminal HTH DNA-binding domain (residues 1–80) and C-terminal portion of unknown function that seems to be conformed of two domains connected by a region of low complexity (residues 167–179) (*Figure 4—figure supplement 1* and *Supplementary file 2*; [*Nyíri et al., 2015*]). Thus, we generated Stl deletional variants lacking the N-terminal DNA binding domain (residues 1–86; StlΔ$^{HTH}$) or the most C-terminal subdomain (residues 176–267; StlΔ$^{Cter}$) (*Figure 4—figure supplement 1*). Unfortunately, these mutants couldn't be analysed in vivo, since the generated Stl mutant repressors had lost the capacity to block the SaPI cycle. To solve that problem, we expressed the different Stl mutants in *E. coli*, and analysed in vitro their capacity to interact with the different Duts. Interestingly, deletion of the N-terminal DNA-binding domain abolished the interaction with the trimeric φ11 but not with the dimeric φO11 Dut. Conversely, the elimination of the C-terminal subdomain impairs the binding to the dimeric but not to the trimeric Dut (*Figure 4A*). Moreover, it has been shown the interaction with the Stl repressor inhibits the dUTPase activity of both dimeric and trimeric Duts (*Hill and Dokland, 2016*; *Szabó et al., 2014*). Here we have confirmed this inhibitory activity for the φ11 and φO11 Duts with the full-length Stl protein (*Figure 4B*). Furthermore, and in agreement with the binding capacity shown by the Stl deletional variants, StlΔ$^{HTH}$ inhibits the dUTPase activity of dimeric but not trimeric Duts, while StlΔ$^{Cter}$ has the opposite capacity (*Figure 4B*). The fact that the SaPIbov1 Stl has particular regions for interacting with the trimeric and dimeric Duts supports the idea that the SaPIbov1 Stl repressor has evolved distinct ways to specifically interact with the dimeric or trimeric Duts.

## The phage 80α encoded Sak recombinase is the inducer for SaPI2

We next addressed the question of whether the previous phenomenon was exclusive to SaPIbov1. To do that, we initially tried to identify the phage 80α inducer for SaPI2, a SaPI frequently responsible for the clinically relevant menstrual toxic shock syndrome (TSS; *Subedi et al., 2007*). Since SaPIs severely interfere with helper phage reproduction, a classical strategy used to identify non-essential SaPI inducers is to generate spontaneous phage mutants that are able to form plaques in the presence of the SaPIs. This strategy selects for phage mutants that have lost the ability to mobilise the islands because of mutations they carry in the SaPI inducer genes. These mutations usually generate non-functional proteins that have also lost their capacity to relieve Stl-mediated repression (*Frígols et al., 2015*; *Tormo-Más et al., 2010*). After many attempts, we obtained only a single spontaneous 80α phage mutant which was able to form plaques on *S. aureus* strain RN4220 containing SaPI2, suggesting that the SaPI2 inducer is absolutely essential for the phage cycle even in laboratory conditions. In this mutant the 3' region of the 80α ORF16 has been lost. Translation of this mutated gene generates a chimeric protein fused with the single strand binding protein (Ssb; 80α ORF17; *Figure 5—figure supplement 1*). Since in this mutant phage the *ssb* gene (including its ribosomal binding site) is unaffected and can be transcribed and translated independently of the chimeric structure, this result suggests that ORF16 is the SaPI2 inducer.

The 80α ORF16 protein belongs to the Sak family of single strand annealing proteins (SSAP, also called recombinases) involved in homologous recombination (*Lopes et al., 2010*; *Scaltriti et al., 2011*). Although for many of these proteins their role in the phage cycle has not been established yet, we have recently demonstrated that this protein is essential for 80α phage replication (*Neamah et al., 2017*). Note, however, that the chimeric Sak-Ssb protein is still functional for the

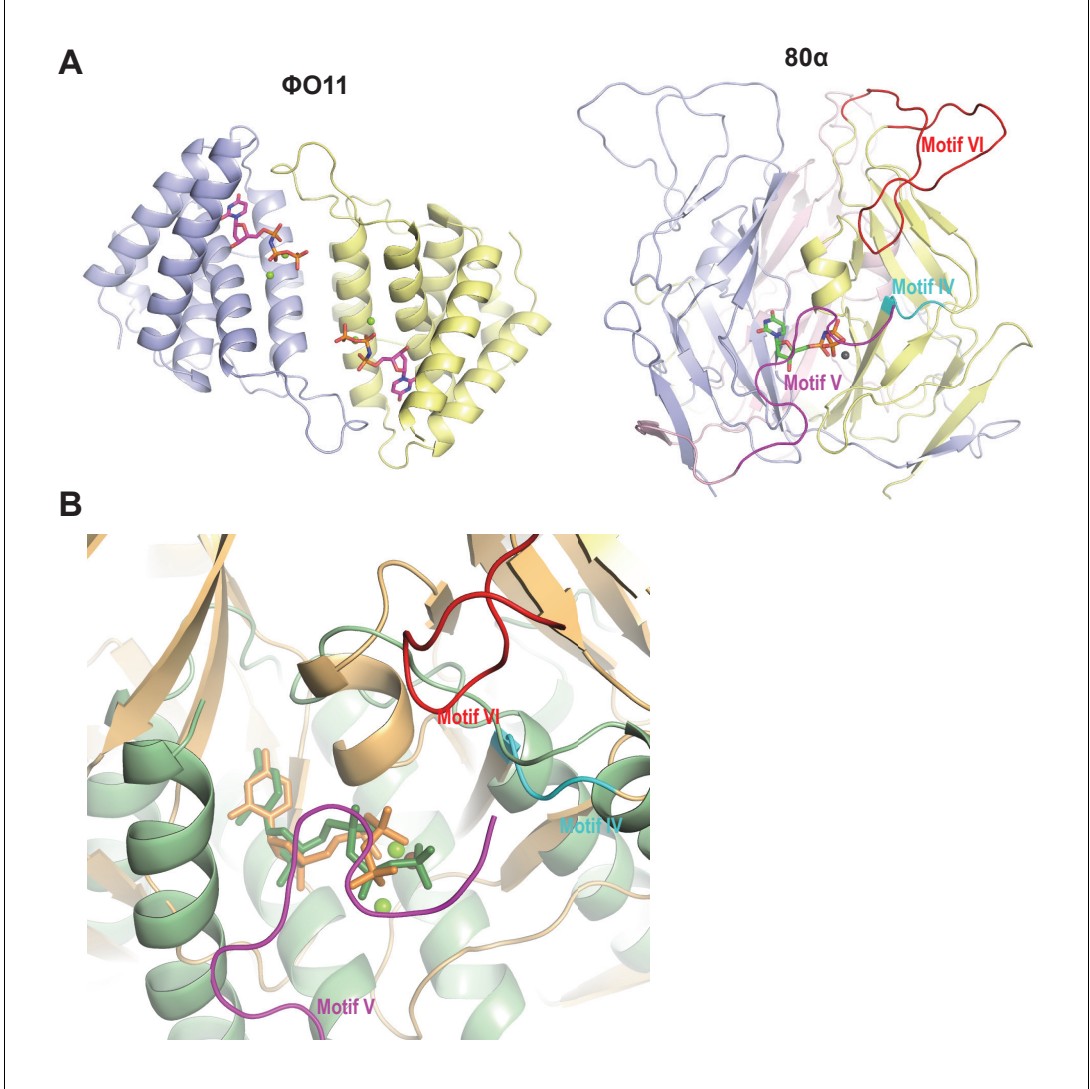

**Figure 3.** Dimeric and trimeric *S. aureus* phagic Duts present completely different folding. (**A**) Cartoon representation of φO11 dimeric Dut (protomers coloured in blue and yellow) and 80α trimeric Dut (PDB 3ZEZ; protomers coloured in blue, yellow and pink) showing the difference in folding between dimeric (all-alpha) and trimeric (all-beta) Duts. The dUPNPP molecules in the active centres are represented in stick and the Mg ions as spheres. For clarity only one dUPNPP molecule is shown in the trimeric structure. The structural motifs implicated in Stl recognition for trimeric Duts (*Maiques et al., 2016*; *Tormo-Más et al., 2013*) are labelled and coloured in cyan, magenta and red for motif IV, V and VI, respectively. (**B**) Superimposition of dUPNPP molecules in the active centres of φO11 (green tones) and 80α (orange tones) shows that the bound nucleotide molecules acquire different conformations (stick representation), including the disposition of the Mg ions (sphere representation), and that the spatial arrangement of the structural elements conforming each active centre is essentially different. No structural element equivalent to the Stl binding motifs of 80α (coloured as in A) is observed in φO11.

DOI: https://doi.org/10.7554/eLife.26487.008

phage, as demonstrated by the fact that the mutant phage encoding this protein still replicates and forms plaques in a sensitive recipient strain.

Expression of the 80α *sak* (ORF16) gene (from the P*cad* promoter in expression vector pCN51) in a SaPI2 positive strain demonstrated that Sak is sufficient to induce this SaPI. Thus, when overexpressed, the cloned *sak* (but not the chimeric Sak-Ssb protein) induced SaPI2 excision and replication (*Figure 5*). As the protein levels produced from these constructs are comparable (*Figure 5*), this result clearly shows that although expressed, the chimeric protein has lost its capacity to induce SaPI2. Moreover, and to confirm that the mechanism involving Sak in SaPI2 induction matches with that previously reported for the other SaPIs, we demonstrated that 80α Sak induces expression of

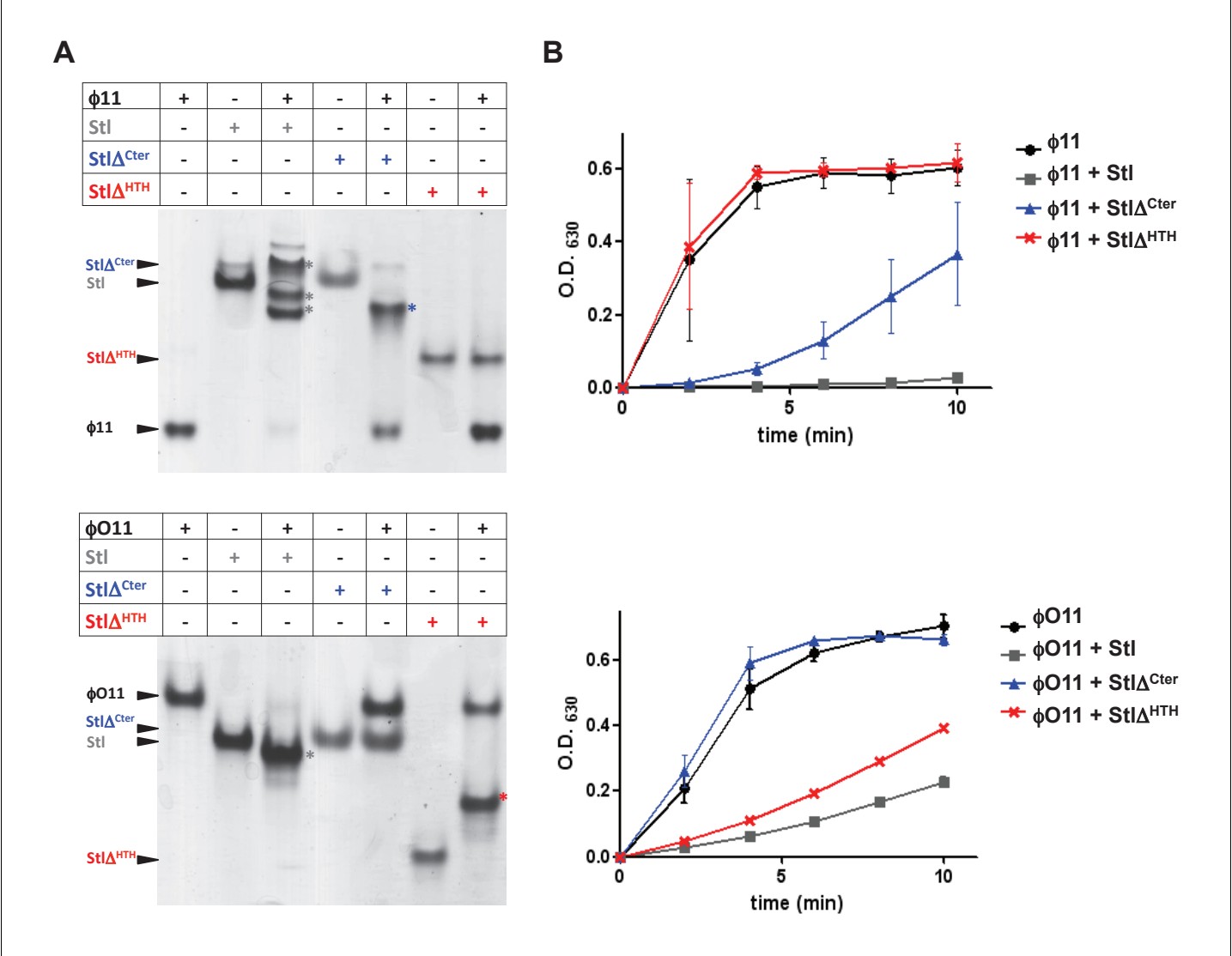

**Figure 4.** SaPIbov1 Stl has different regions to interact with the trimeric and dimeric Duts. (**A**) Native gel mobility shift assays were used to test the binding capacity of φO11 dimeric and φ11 trimeric Duts with full-length and truncated versions of Stl. The appearance of bands with alternated migration with respect to the individual proteins (labelled by asterisk) indicates formation of a complex. (**B**) dUTPase activity for φO11 and φ11 was measured by malachite green assay in the presence and the absence of Stl variants. The reaction time-course is represented as the development of green colour (measured at 630 nm). Results are representative of three independent experiments.
DOI: https://doi.org/10.7554/eLife.26487.009

The following figure supplement is available for figure 4:

**Figure supplement 1.** Stl models and constructs design.
DOI: https://doi.org/10.7554/eLife.26487.010

the SaPI2 Stl repressed genes. This was confirmed using plasmid pJP1977, which carries a β-lactamase reporter gene fused to *xis*, downstream of *str* and the Stl$_{SaPI2}$-repressed *str* promoter, and also encodes Stl$_{SaPI2}$ (see *Figure 6A*). The cloned *sak* gene was introduced on vector pCN51 and expression was tested in the presence or absence of an inducing concentration of CdCl$_2$. Induction of *sak*, but not the chimeric *sak-ssb*, strongly increased β-lactamase expression from the *str* promoter (*Figure 6A*). Moreover, the predicted protein–protein interaction between Sak and the Stl$_{SaPI2}$ repressor was confirmed by co-expression and affinity purification of His$_6$-Stl$_{SaPI2}$ and untagged Sak proteins. It was possible to co-purify a complex between His$_6$-Stl$_{SaPI2}$ and Sak (*Figure 6B*), whereas untagged Sak alone did not bind to the resin. The chimeric Sak-Ssb, which does not derepress SaPI2

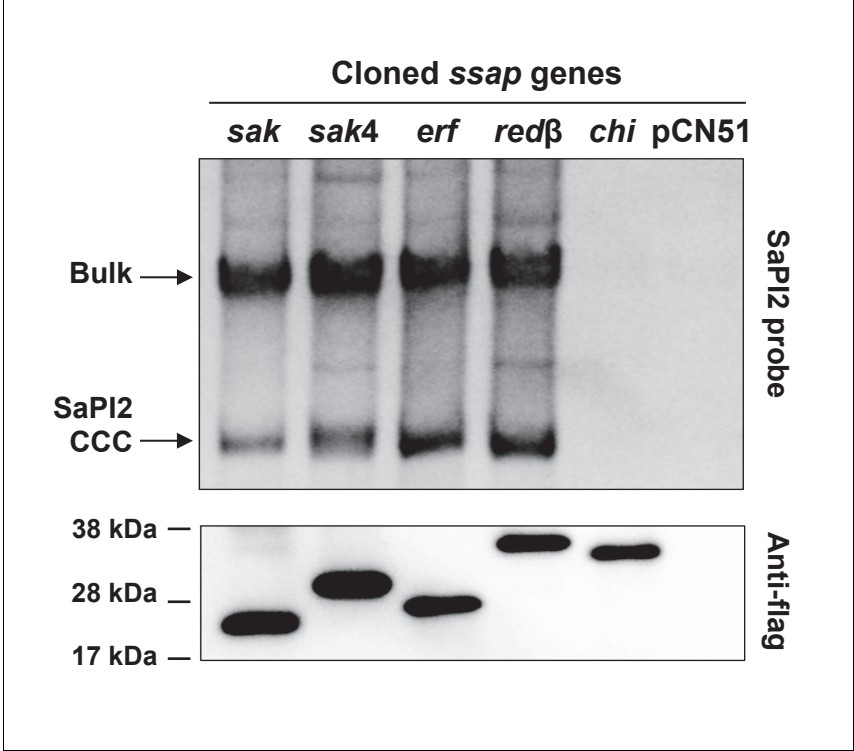

**Figure 5.** Induction of SaPI2 by different phage-encoded SSAPs (recombinases). A non-lysogenic derivative of strain RN4220 Δ*spa* carrying SaPI2 was complemented with plasmids expressing different 3xFLAG-tagged SSAP proteins. One millilitre of each culture (optical density (OD)$_{540nm}$=0.3) was collected 3 hr after treatment with 5 µM CdCl$_2$ and used to prepare standard mini-lysates, which were resolved on a 0.7% agarose gel, Southern blotted and probed for SaPI2 DNA. The upper band is 'bulk' DNA, including chromosomal, phage, and replicating SaPI. CCC indicates covalently closed circular SaPI2 DNA. In these experiments, as no helper phage was present, the excised and replicating SaPI DNA appears as part of the bulk DNA or as CCC molecules, rather than the linear monomers that are seen following helper phage-mediated induction and packaging. The lower panel is a western blot probed with antibody to the FLAG-tag carried by the SSAP proteins. *sak*: 80α ORF16; *sak4*: φ52A ORF16; *erf*: φSLT ORF17; *redβ*: φN315 ORF SA1794; *chi*: chimeric 80α *sak-ssb*; pCN51: empty vector.

DOI: https://doi.org/10.7554/eLife.26487.011

The following figure supplements are available for figure 5:

**Figure supplement 1.** Sequence of the chimeric ORF16-17 protein.
DOI: https://doi.org/10.7554/eLife.26487.012
**Figure supplement 2.** Localisation of the recombinase and *ssb* genes in different staphylococcal phage genomes.
DOI: https://doi.org/10.7554/eLife.26487.013
**Figure supplement 3.** Alignment of predicted Sak (80α) and Sak4 (φ52A) staphylococcal phage SSAPs.
DOI: https://doi.org/10.7554/eLife.26487.014
**Figure supplement 4.** Alignment of predicted staphylococcal phage SSAPs.
DOI: https://doi.org/10.7554/eLife.26487.015

(*Figure 6A*), did not co-purify with His$_6$-Stl$_{SaPI2}$, confirming the specificity of the His$_6$-Stl$_{SaPI2}$::Sak interaction. The identity of each of these bands was confirmed by amino acid sequencing and mass spectrometry.

## The Sak4 recombinase also induces SaPI2

Next, and based on the fact that both dimeric and trimeric Duts induce SaPIbov1, we explored the possibility that the SaPI2 Stl repressor could also target different phage proteins, significantly increasing the capacity of SaPI2 to be induced and transferred. Interestingly, phages φ80 and φ52A can also induce the SaPI2 cycle (*Ram et al., 2014*), although none of them encodes a 80α Sak protein. To test the possibility that SaPI2 was targeting another protein, we tried to identify the SaPI2

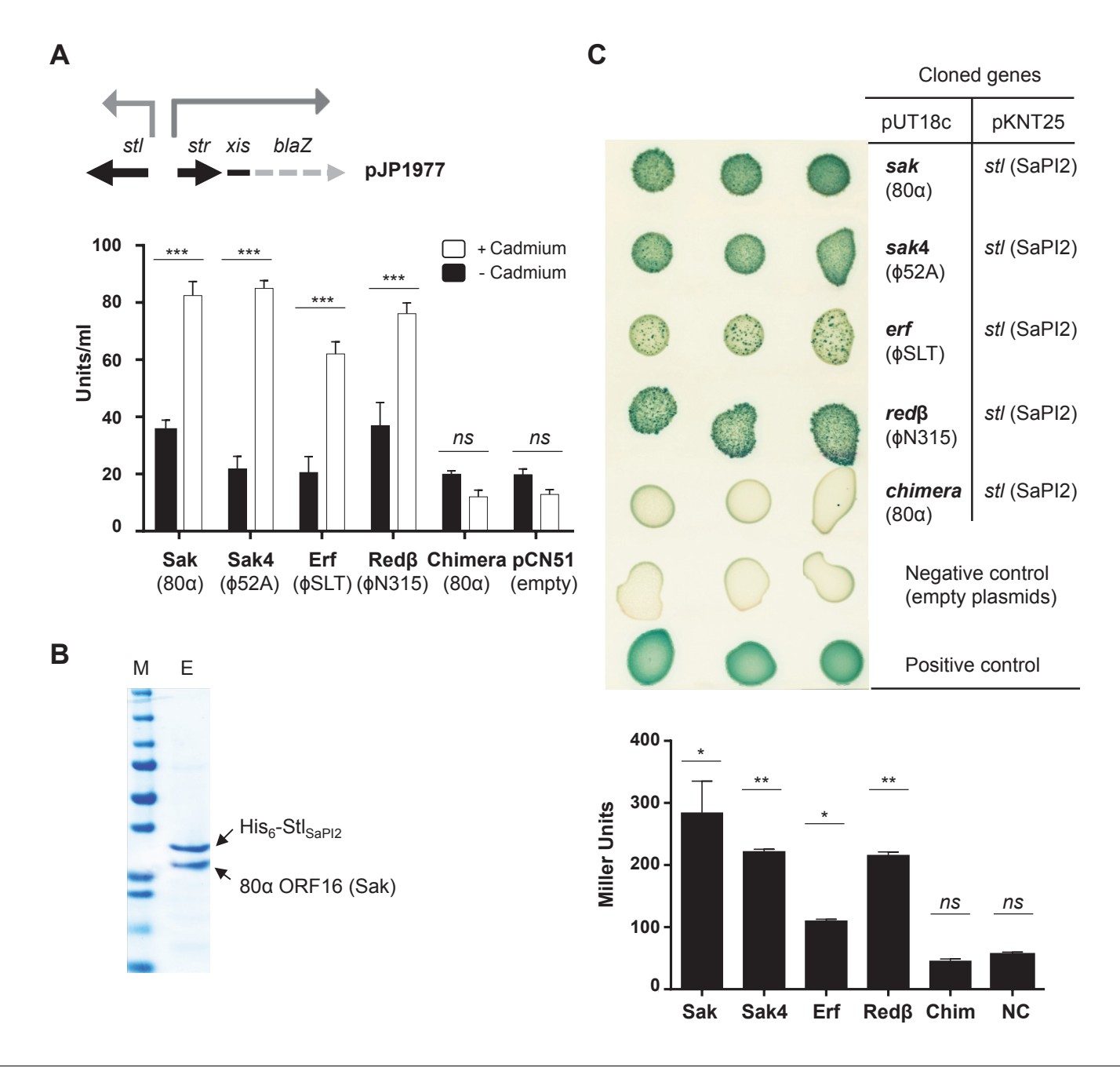

**Figure 6.** Phage SSAPs bind SaPI2 Stl protein. (**A**) Derepression of *str* transcription by *ssap* expression. Top, schematic representation of the *blaZ* transcriptional fusion generated in plasmid pJP1977. Bottom, strains containing pJP1977- and pCN51-derivative plasmids expressing the different SSAPs under study were assayed for β-lactamase activity in the absence of or 3 hr after induction with 5 μM $CdCl_2$. Samples were normalized for total cell mass. Experiment data is in triplicate. Error bars represent SEM. A 2-way ANOVA with Sidak's multiple comparisons test was performed to compare mean differences within rows. Adjusted *p* values were as follows: Sak = 0.0001[***], Sak4 = 0.0001[***], Erf = 0.0001[***], Redβ=0.0001[***], chimera = 0.999[ns]. *ns*, not significant. (**B**) Affinity chromatography of 80α Sak (ORF16) using $His_6$–$Stl_{SaPI2}$. *E. coli* strain expressing the pair was isopropyl-β-D-thiogalactoside (IPTG)-induced and, after disruption of the cells, the expressed proteins were applied to a $Ni^{2+}$ agarose column and eluted. The presence of the different proteins was monitored in the elute fraction (E) by Coomassie staining. M: molecular weight marker. (**C**) Bacterial adenylate cyclase-based two-hybrid (BACTH) analysis. Spots in each row represent three independent colonies. Plasmid combinations are indicated in the right columns. Bottom, quantification of the BACTH analysis after 2 hr of IPTG (5 mM) induction. Experiment data is in triplicate. Error bars represent SEM. A 1-way ANOVA with Sidak's multiple comparisons test was performed to compare mean differences within rows. Adjusted *p* values were as follows; Sak = 0.0221*, Sak4 = 0.0030**, Erf = 0.0158*, Redβ=0.0014**, chimera (Chim) = 0.1980[ns]. *ns*, not significant.

*Figure 6 continued on next page*

*Figure 6 continued*

DOI: https://doi.org/10.7554/eLife.26487.016

The following source data is available for figure 6:

**Source data 1.** β-lactamase assay data and statistical analysis for the recombinases.
DOI: https://doi.org/10.7554/eLife.26487.017
**Source data 2.** BACTH analysis data and statistical analysis for the recombinases.
DOI: https://doi.org/10.7554/eLife.26487.018

inducer in phages ϕ80 and ϕ52A by generating spontaneous phage mutants that can grow in the presence of the island. After many attempts, we did not get any phage mutants capable of forming plaques in a SaPI2 positive strain, suggesting that the SaPI2 inducers are also essential for the biology of these phages, even in laboratory conditions. In view of this result, and bearing in mind that both the dimeric and trimeric Duts have the same biological (enzymatic) function for the phage, we hypothesised that the ϕ80 or ϕ52A SaPI2 inducers would be functionally related to the 80α Sak protein. Since the *S. aureus* phages display synteny, we speculated that the genes located in the same genome position as the 80α *sak* gene would be essential for the phage, would have a recombinase function, and would encode for the SaPI2 inducer. While phages ϕ80 and ϕ52A do not contain an *orf* homologue to 80α *sak*, all three phages encode identifiable *ssb* genes, which in the case of the 80α phage is located downstream of the *sak* gene (*Figure 5—figure supplement 2*). Thus, we analysed the possibility that the genes upstream of *ssb* were the SaPI2 inducers. Both ϕ80 and ϕ52A phages carried an identical gene, named ORF13 in phage ϕ80 and ORF16 in phage ϕ52A, which encodes a non-related protein to the 80α Sak (*Figure 5—figure supplement 3*). This protein belongs to a distinct family of SSAPs, Sak4 (*Lopes et al., 2010*). While Sak4 and Sak are not homologous in sequence (*Figure 5—figure supplement 3*), we have recently demonstrated that they are both SSAPs (recombinases) performing a similar function in their cognate phages (*Neamah et al., 2017*).

The results above support the hypothesis proposing that unrelated proteins performing the same function for the phages could all act as inducers for a specific SaPI. Thus, expression of the ϕ80 and ϕ52A Sak4 proteins in a SaPI2 positive strain demonstrated that they are sufficient to induce the SaPI2 cycle (*Figure 5*). Moreover, expression of the s*ak4* genes strongly increased β-lactamase expression from the Stl-repressed *str* promoter (*Figure 6A*). Since expression of the ϕ52A Sak4 protein in *E. coli* generated an insoluble protein which aggregates, we couldn't co-purify a complex between His$_6$-Stl$_{SaPI2}$ and untagged ϕ52A Sak4. However, a two-hybrid assay confirmed the strong interaction between both the ϕ52A Sak4 recombinase and the SaPI2 Stl repressor and between the 80α Sak protein and the SaPI2 Stl pair (used here as a control; *Figure 6C*), confirming that the phage Sak4 protein is a *bona fide* SaPI2 inducer.

## Unrelated phage-encoded recombinases induce SaPI2

Since SaPI2 superimposes a high cost for the phage, it could be possible that staphylococcal phages would initially avoid this interference by encoding additional SSAPs, unrelated to Sak or Sak4. In turn, and if the hypothesis we propose here is correct, it could also be possible that the SaPI repressor would evolve to target these new phage encoded recombinase proteins. In silico scrutiny looking at the genes located upstream of the *ssb* genes revealed that staphylococcal phages encode at least 4 distant SSAP families, including Erf, Redβ, and the aforementioned Sak and Sak4 (*Supplementary file 3*). All the staphylococcal phages encode one SSAP, in accordance with the fact that these proteins are essential for the phage (*Neamah et al., 2017*). To test the possibility that these other unrelated recombinases also induced SaPI2, we characterised in detail those present in phages ϕSLT (ORF 17) and ϕN315 (SA1794), which belong to the Erf and Redβ families of SSAPs, respectively, and have completely different sequences (*Figure 5—figure supplement 4*). We selected phage ϕSLT because it is clinically relevant, encoding the Panton-Valentine leukocidin (PVL) toxin. Applying the same methodology and strategies previously used to characterise Sak and Sak4, our results confirm that: (i) the expression of the ϕSLT Erf and ϕN315 Redβ proteins is sufficient to induce the SaPI2 cycle (*Figure 5*); (ii) expression of these recombinases prevents Stl from binding to the SaPI2 *stl-str* divergent region (*Figure 6A*) and (iii) the two-hybrid assay confirmed the interaction between the SaPI2 Stl repressor and the Erf and Redβ recombinases (*Figure 6C*).

Finally, since the existence of different interacting domains in the Stl repressor explains why both the dimeric and trimeric proteins can induce SaPIbov1, we wondered if a similar mechanism was employed by the SaPI2 island. Structure-based modelling of Sak, Sak4, Erf and Redβ suggested they are unrelated, although Sak, Erf and Redβ can be connected through remote homology relationships (*Lopes et al., 2010*). Thus, it has been proposed that Sak, Erf and Redβ belong to a large superfamily adopting a shortcut Rad52-like fold (*Lopes et al., 2010*). However, structural models produced with I-Tasser (*Yang et al., 2015*) and Phyre2 (*Kelley et al., 2015*) servers for Sak (phage 80α), Erf (φSLT) and Redβ (φN315) only proposed the Rad52 fold for Sak, whereas for Erf and Redβ alternating foldings, non-related with Rad52 recombinases, were proposed with low confidence (*Figure 7* and *Supplementary files 4A and B*). By contrast, remote homologs to Sak4 are predicted to adopt a shortcut Rad51/RecA fold (*Lopes et al., 2010*) and models obtained from I-Tasser and Phyre2 servers proposed this fold for the φ52A Sak4 recombinase with good confidence (*Figure 7* and *Supplementary files 4A and B*). Taken together, these results suggest that the most likely scenario explaining why the SaPI2 Stl repressor can interact with different recombinases is the existence of different interacting domains in the repressor.

## Pirating conserved phage processes supports inter-specific SaPI transfer

Both we and others have previously demonstrated that the SaPIs can be inter- and intra-generically transferred (*Chen and Novick, 2009*; *Chen et al., 2015*; *Maiques et al., 2007*). Although this process occurs at astonishingly high frequencies in the lab, its impact in nature remains unsolved. The fact that the mechanisms involved in the life cycle of the phages are conserved among species raised an interesting possibility: by targeting conserved phage processes SaPI-like elements would be successfully spread and maintained in nature. To test this hypothesis, we searched for SaPIbov1 and SaPI2 Stl homologs in the database. Different SaPIbov1 Stl homologs were identified in PICI elements present in *S. aureus*, *Staphylococcus hominis*, *Staphylococcus haemolyticus*, *Staphylococcus lugdunensis*, *Staphylococcus saprophyticus* and *Staphylococcus simulans*. SaPI2 Stl homologs were also identified in many different Staphylococci, including *Staphylococcus argenteus*, *Staphylococcus caprae*, *S. lugdunensis*, *Staphylococcus epidermidis*, *S. haemolyticus*, *S. simulans*, *Staphylococcus xylosus* and *Staphylococcus capitis*, as well as in PICI from other Gram-positive bacteria, including *Bacillus decisifrondis* or *Streptococcus pyogenes*. *Supplementary files 5A and B* delineate characteristics of the different PICI elements and the identity among the different Stl repressors encoded by the Staphylococci PICIs, respectively. Of note is the fact that some islands, present in different species, encode identical Stl repressors, suggesting inter-species transfer. This was the case for SaPIbov1 (*S. aureus*) and SlCIVISLISI_25 (*S. lugdunensis*), both encoding the SaPIbov1 repressor, and SaPI2 (*S. aureus*), SarCISJTUF21285 (*S. argenteus*), ScCIM23864:W1 (*S. caprae*) and SlCIFDAAR-GOS_141 (*S. lugdunensis*), all encoding the SaPI2 Stl.

To test if these Stl repressors interact with the SaPIbov1 or SaPI2 inducers, we used the aforementioned strategy to generate a set of plasmids in which the divergent *str/str-xis* region of the PICIs was fused to a β-lactamase reporter gene. These derivatives were generated for the PICIs encoding the most distantly related Stl repressors: ShoCI794_SEPI (*S. hominis*) and ShaCI51-48 (*S. haemolyticus*), encoding a SaPIbov1 Stl homolog, and ShaCI137133 (*S. haemolyticus*), SeCI-NIHLM095 (*S. epidermidis*) and SsCIUMC-CNS-990 (*S. simulans*) carrying a SaPI2 Stl homolog. Next, the capacity of the dimeric φO11 or trimeric φ11 Duts (for the SaPIbov1-like Stl repressors), or the ability of the different SSAPs (for the SaPI2-like Stl repressors) to induce the PICI cycle was tested by introducing the pCN51 derivatives expressing the different SaPI inducers in the strains carrying the reporter plasmids. Remarkably, both the dimeric φO11 and trimeric φ11 Duts induced β-lactamase expression from the Stl-repressed *str* promoters present in the ShoCI794_SEPI and ShaCI51-48 islands (*Table 1*), suggesting that the Stl repressors encoded in all these islands have a common origin.

Even more interesting were the results obtained with the SaPI2 Stl homologs (*Table 2*). All the islands were induced by at least one of the recombinases, although the distribution was not as uniform as with the Duts. Thus, the 80α coded Sak induced SaPI2, SeCINIHLM095 and ShaCI137133, but not ScCIUMC-CNS-990 (*Table 2*). The chimeric 80α Sak-Ssb protein induced none, supporting that the different PICI coded Stl repressors are structurally related (*Table 2*). Interestingly, the φ52A coded Sak4 recombinase induced SaPI2 and ScCIUMC-CNS-990, while the φSLT Erf recombinase

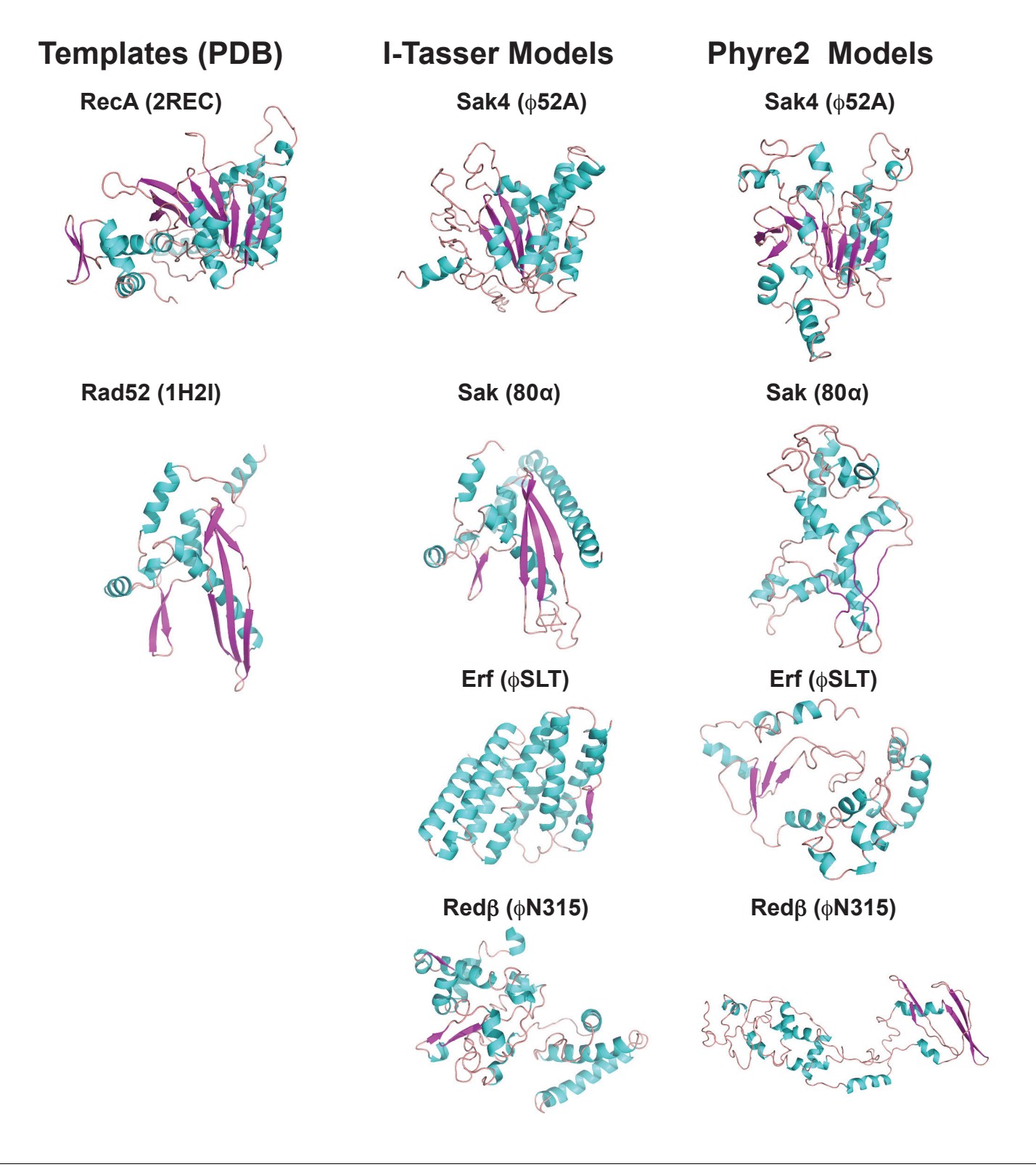

**Figure 7.** 3D models for 80α, φSLT, φ52A and φN315 SSAPs. Cartoon representation of 3D structures of 80α, φSLT, φ52A and φN315 phage recombinases generated by I-Tasser and Phyre2. Alpha helices, beta strands and loops are coloured in cyan, magenta and orange, respectively. The experimental structures of RecA from *E. coli* (PDB 2REC) proposed as structural homolog of the φ52A recombinase, and the human Rad52 recombinases (1H2I) proposed as structural homolog of 80α, φSLT, and φN315 recombinases are also presented for folding comparison.

*Figure 7 continued on next page*

*Figure 7 continued*

DOI: https://doi.org/10.7554/eLife.26487.019

induced SaPI2 and SeCINIHLM095. Finally, the ϕN315 coded Redβ recombinase uniquely induced SaPI2, but not the other islands (*Table 2*). Taken together, including the previous results with the SaPIbov1 Stl mutants, these results strongly support the idea that although originally related, the different Stl repressors have evolved different domains to interact with the phage-coded inducers.

It is striking how the SaPIs have evolved an elegant tactic to be highly transferred both intra- and inter-generically. However, and in the case of the inter-generic transfer of the elements, to be completely effective this strategy requires that the phages infecting the new SaPI-recipient species encode the conserved SaPI inducers. To test this, we analysed the presence of SaPIbov1 or SaPI2 inducing genes in a subset of staphylococcal phages infecting species other than *S. aureus*. As shown in *Supplementary file 6*, we were able to identify homologs to the previously characterised SaPIbov1 or SaPI2 inducer in all the analysed phages, although with different degrees of identity among the members of the distinct families. Next, and to support the idea that once the inter-species transfer occurs the PICI can be maintained in the new recipient species, we tested whether the Dut encoded in the *S. epidermidis* phage IPLA6, or the recombinase encoded in *S. epidermidis* phage PH15, were capable of inducing the cycle of the different PICI elements encoding SaPIbov1 or SaPI2 Stl homologs. That was the case, and the behaviour of the *S. epidermidis* IPLA6 trimeric Dut was identical to that observed for the trimeric ϕ11 Dut (*Table 1*), while the functioning of the ϕPH15 Sak recombinase was indistinguishable from that observed with the homologue *S. aureus* 80α coded Sak (*Table 2*). In summary, our results confirm the idea that the PICIs have established a fascinating parasitic strategy that may allow their promiscuous transfer and widespread maintenance in nature.

## Inter-species PICI transfer occurs in nature

The fact that some islands present in different species encode identical proteins (including not just the Stl repressors as demonstrated here but also other PICI proteins) strongly supports the idea that

**Table 1.** Dimeric and trimeric dUTPases induce PICIs from other species encoding SaPIbov1-like Stl repressors[a].

| | Phage Duts | | | |
| | Dimeric | | Trimeric | |
| | ϕNM1 *S. aureus* | ϕO11 *S. aureus* | ϕIPLA6 *S. epidermidis* | ϕ11 *S. aureus* |
|---|---|---|---|---|
| **PICIs** | | | | |
| SaPIbov1 | ++ | ++ | ++ | +++ |
| ShoCI794_SEPI | ++ | ++ | ++ | +++ |
| ShaCI51-48 | + | + | + | + |

[a]Results are from 5 independent β-lactamase/Nitrocefin assay experiments, using the dual plasmid system described in the text. Levels of induction were based on the calculated units/ml with the following ranges: (-): no induction, <1 Units/ml; (+): low induction, 1–5 Units/ml; (++): moderate induction, 5–10 Units/ml; (+++): high induction, >10 Units/ml. A 2-way ANOVA with Sidak's multiple comparisons test was performed to compare mean differences within rows. The differences observed among the +,++or +++ samples, compared with the controls or the - samples, were in all cases statistically significant (p<0.05).
DOI: https://doi.org/10.7554/eLife.26487.020

The following source data available for Table 1:

**Source data 1.** β-lactamase assay data and statistical analysis for the SaPIbov1 Stl homologues.
DOI: https://doi.org/10.7554/eLife.26487.021

**Table 2.** Unrelated SSAPs differentially induce PICIs from other species encoding SaPI2-like Stl repressors[a].

| | SSAPs | | | | | |
| | Sak (80α) *S. aureus* | Chimera (80α) *S. aureus* | Sak (PH15) *S. epidermidis* | Sak4 (52A) *S. aureus* | Erf (SLT) *S. aureus* | Redβ (N315) *S. aureus* |
|---|---|---|---|---|---|---|
| **PICIs** | | | | | | |
| SaPI2 | ++ | - | ++ | +++ | ++ | + |
| SeCINIHLM095 | ++ | - | ++ | - | ++ | - |
| ShaCI137133 | ++ | - | ++ | - | - | - |
| ScCIUMC-CNS990 | - | - | - | +++ | - | - |

[a]Results are from 3 independent β-lactamase/Nitrocefin assay experiments, using the dual plasmid system described in the text. Levels of induction were based on the calculated units/ml with the following ranges: (-): no induction, <1 Units/ml; (+): low induction, 1–5 Units/ml; (++): moderate induction, 5–10 Units/ml; (+++): high induction,>10 Units/ml. A 2-way ANOVA with Sidak's multiple comparisons test was performed to compare mean differences within rows. The differences observed among the +,++or +++ samples, compared with the controls or the - samples, were in all cases statistically significant ($p<0.05$).
DOI: https://doi.org/10.7554/eLife.26487.022

The following source data available for Table 2:

Source data 1. β-lactamase assay data and statistical analysis for the SaPI2 Stl homologues.
DOI: https://doi.org/10.7554/eLife.26487.023

some ancestral elements were horizontally transferred among species. In the different species these elements probably evolved independently, trying to adapt to the new cognate host. Our previous results, however, suggest that by pirating conserved phage mechanisms this inter-species transit probably occurs constantly in nature. To test that possibility, we scrutinised the genome databases looking for identical PICIs present in different staphylococcal species. We initiated this analysis by comparing the genomes of those PICIs encoding identical Stl proteins. Highlighting the versatility of these elements and the successful strategy they use to spread in nature, our analysis revealed that the ScCIM23864:W1 (*S. caprae*) and SlCIFDAARGOS_141 (*S. lugdunensis*) elements are identical (just 3 mismatches over 13,847 nt; *Supplementary file 7*).

## Discussion

The manner by which related SaPIs have acquired the ability to exploit conserved phage processes by targeting structurally unrelated proteins as antirepressors represents a remarkable evolutionary adaptation. Our results suggest that the most likely scenario explaining why the SaPI/PICI Stl repressors can interact with different phage coded inducers is the existence of different interacting domains in the SaPI Stl repressors. The presence of these different domains highlights the co-evolutionary and constant battle established between the helper phages, trying to avoid PICIs induction, and the parasitic PICIs, trying to interact with non-inducing phages (*Frígols et al., 2015*). This mechanism could also be responsible, at least in part, for the widespread distribution of PICIs in nature. Note that we have recently demonstrated the existence of these elements in many Gram-positive cocci (*Martínez-Rubio et al., 2017*).

We hypothesised that at the beginning of the SaPI-phage war, a single phage protein may have been originally targeted; to escape from SaPI de-repression, because SaPIs interfere with phage maturation, substitution of the gene encoding this protein to one expressing a non-related, but functionally similar protein could have had a selective advantage for the phage. A second stage in SaPI evolution could have involved divergence of the SaPI repressor, enabling it to complex with structurally non-related phage proteins. The fact that the Stl repressors interact with structurally unrelated proteins performing the same function makes this strategy unique in nature and extremely effective. Note that in terms of increasing their transferability, a more simple strategy for the SaPIs could have been to select for Stl repressors that can interact with proteins performing different functions for the

phage. However, since phages have mosaicism, encoding multiple versions of unrelated proteins performing the same function (as also demonstrated here), this strategy would select for phages insensitive to the SaPIs that encode the correct combination of non-inducing proteins. By contrast, and since the processes targeted by the SaPIs are extremely well conserved in the staphylococcal phages, the fact that the SaPIs target different versions of proteins involved in the same biological processes limits the capacity of the phages to overcome SaPI parasitism, ensuring the transferability of these elements. Thus, our results show that SaPI-phage interactions represent a remarkable microcosm within the bacterial intracellular universe, highlighting SaPIs as one of the most fascinating and effective subcellular parasites.

However, our results raise an interesting question. Why do some repressors interact just with one inducer, limiting their capacity to be transferred, while others seem to have a broader spectrum of inducers? Our hypothesis is that although all the analysed phages encode putative SaPI inducers, these are different in sequence (see *Supplementary file 6*), so the repressors present in the different PICIs have evolved to increase their interaction with the specific inducers encoded in the cognate phages infecting these bacterial species. This also would explain the divergence in sequence observed in related Stl repressors. This hypothesis is currently under study.

*Lactococcus lactis* encodes a plasmid with an abortive infection mechanism, AbiK (*Bouchard and Moineau, 2004*). As occurs with SaPI2, the proteins targeted by the AbiK system are the different phage encoded SSAPs involved in homologous recombination (*Bouchard and Moineau, 2004*). Although the mechanism by which AbiK blocks phage reproduction remains unclear, it does not seem to involve the formation of a complex between the AbiK protein and the recombinases, as occurs with SaPI2 (*Bouchard and Moineau, 2004*; *Wang et al., 2011*).

Since the discovery of the SaPIs, it has gradually become apparent that prophages and PICIs have evolved in much more interesting ways than has generally been realised. PICIs are sophisticated, elegant and extremely effective parasites. They have incorporated an impressive arsenal of effective strategies to interfere with helper phages, ensuring their presence in nature (*Penadés and Christie, 2015*). We anticipate here that novel and unexpected mechanisms of PICI-mediated phage interference will soon be characterised, which will highlight the fascinating biology of these subcellular creatures and their cognate helper phages.

## Materials and methods

### Bacterial strains and growth conditions

The bacterial strains used in this study are listed in *Supplementary file 8A*. *S. aureus* was grown in Tryptic soy broth (TSB) or on Tryptic soy agar plates. *E. coli* was grown in LB broth or on LB agar plates. Antibiotic selection was used where appropriate. Preparation and analysis of phage lysates was performed essentially as previously described (*Ubeda et al., 2008*).

### DNA methods

General DNA manipulations were performed using standard procedures. Plasmid constructs used in this study (*Supplementary file 8B*) were generated by cloning PCR products obtained with oligonucleotide primers, listed in *Supplementary file 8C*.

Detection probes for SaPI DNA in Southern blots were generated by PCR using primers SaPI-bov1-112mE and SaPIbov1-113cB (SaPIbov1 and SaPIbov5) or Tet-1m and Tet-2c (SaPI2), listed in *Supplementary file 8C*. Probe labelling and DNA hybridization were performed following the protocol provided with the PCR-DIG DNA-labelling and chemiluminescent detection kit (Roche). Southern blot experiments were performed as previously described (*Tormo-Más et al., 2010*).

$\phi$O11 *dut* was cloned into pET28a vector (Novagen) using primers listed in *Supplementary file 8C*. Plasmids pETNKI-Stl$\Delta^{HTH}$ and pETNKI-Stl$\Delta^{Cter}$ for expression of Stl deletional variants were produced using plasmid pETNKI-Stl as template (*Maiques et al., 2016*). pETNKI-Stl$\Delta^{Cter}$ plasmid expressing Stl residues from 1 to 176 was generated by site direct mutagenesis introducing a stop codon in pETNKI-Stl after Lys176 using the Stl_M1-K176_Fw and Stl_M1-K176_Rv primers and Q5 Site-Directed Mutagenesis Kit (NEB). pETNKI-Stl$\Delta^{HTH}$ plasmid expressing Stl residues from 87 to 267 was generated by PCR-amplifying the encoding region with the primers Stl_T87-N267_Fw and Stl_T87-N267_Rv. The Ligation-Independent Cloning (LIC) system (*Savitsky et al., 2010*) was used

to clone the PCR fragment into the pETNKI-his-SUMO3-LIC plasmid (kindly supplied by Patrick Celie, NKI Protein facility) previously digested with a *Kpn*I (Fermentas). All clones were sequenced at the IBV Core Sequencing facility or by Eurofins MWG Operon.

## Southern and western blot sample preparation

Samples were taken at times 0' and 3 hr following plasmid induction and pelleted. The samples were re-suspended in 50 µl lysis buffer (47.5 µl TES-Sucrose and 2.5 µl lysostaphin [12.5 µg ml$^{-1}$]) and incubated at 37°C for 1 hr. For the Southern blot analysis, 55 µl of SDS 2% proteinase K buffer (47.25 µl H2O, 5.25 µl SDS 20%, 2.5 µl proteinase K [20 mg ml$^{-1}$]) was added before incubation at 55°C for 30 min. Samples were vortexed for 1 hr with 11 µl of 10x loading dye. Cycles of incubation in dry ice and ethanol, then at 65°C were performed. Samples were run on 0.7% agarose gel at 25V overnight. DNA was transferred to a membrane and exposed using a DIG-labelled probe and anti-DIG antibody, before washing and visualisation.

Preparation of *S. aureus* samples for western blot was performed by re-suspending pellets in 200 µl digestion/lysis buffer (50 mM Tris-HCl, 20 mM MgCl2, 30% w/v raffinose) plus 1 µl of lysostaphin (12.5 µg ml$^{-1}$), mixed briefly, and incubated at 37°C for 1 hr. 2X Laemmli sample buffer (Bio-Rad, 2-mercaptoethanol added) was added to the samples, which were heated at 95°C for 10 min, put on ice for 5 min and fast touch centrifuged. Samples were run on SDS-PAGE gels (15% Acrylamide, Bio-Rad 30% Acrylamide/Bis Solution) before transferring to a PVDF transfer membrane (Thermo Scientific, 0.2 µM) using standard methods. Western blot assays were performed using anti-Flag antibody probes (Monoclonal ANTI-FLAG M2-Peroxidase (HRP) antibody, Sigma-Aldrich) as per the protocol supplied by the manufacturer.

## Two-hybrid assay

The two-hybrid assay for protein-protein interaction was done as described previously (*Battesti and Bouveret, 2012*) using two compatible plasmids; pUT18c expressing T18 fusion with the individual recombinases, and pKNT25 expressing the T25 fusion with the Stl$_{SaPI2}$. Both plasmids were co-transformed into *E. coli* BTH101 for the Bacterial Adenylate Cyclase Two Hybrid (BACTH) system and plated on LB +Ampicillin and Kanamycin + X gal as an indicator. After incubation at 30°C for 24 hr (early reaction) and 48 hr (late reaction), the protein-protein interaction was detected by a colour change. Blue colonies represent an interaction between the two clones, while white/yellow colonies are negative for any interaction.

For quantification of the BACTH analysis, overnight cultures were diluted 1/100 and grown to mid-log before induction with 5 mM IPTG. After 2 hr, 2 ml of culture was sampled and pelleted, before resuspension in the same volume of Z buffer (0.06M Na$_2$HPO$_4$.7H$_2$O, 0.04M NaH$_2$PO$_4$.H$_2$O, 0.01M KCl, 0.001M MgSO$_4$, 0.05M β-mercaptoethanol, pH 7.0). The OD$_{600}$ was recorded and cells were permeabilized with chloroform and 0.1% SDS. The assay reaction was started using ONPG (*o*-nitrophenyl-β-D-galactoside, 4 mg/ml), and vortexed and incubated at 30°C until yellow. The reaction was stopped using Na$_2$CO$_3$ and the reaction time recorded. Samples were spun down and the OD$_{420}$ and OD$_{550}$ were recorded. Miller Units were calculated as follows, where *T* is time of reaction (minutes) and *V* is the volume of culture used in the assay (ml): Miller Units = 1000 x (OD$_{420}$ - 1.75 x OD$_{550}$) / (*T* x *V* x OD$_{600}$).

## Enzyme assays

For the β-Lactamase assays, cells were obtained at 0.2–0.3 OD$_{540}$ and at 5 hr post-induction with/without 5 µM CdCl$_2$. β-Lactamase assays, using nitrocefin as substrate, were performed as described (*Tormo-Más et al., 2010*), using a ELx808 microplate reader (BioTek). An adjustment was made in reading time, with plates read every 20 s for 30 mins. β-Lactamase units/ml are defined as *[(slope) (Vd)]/[(Em)(l)(s)]*. *Slope* is the Δabsorbance/hour, *V* is the volume of the reaction, *d* is the dilution factor, *Em* is the millimolar extinction coefficient for the nitrocefin (20,500 M$^{-1}$ cm$^{-1}$ at 486 nm), *l* is the path length (cm), and *s* is the sample amount. dUTPase activity was measured by Malachite Green assay as previously described (*Maiques et al., 2016*). Briefly, Dut (30 nM) alone or in presence of a 5X molar ratio (monomer) of Stl (full length or truncated versions) was incubated overnight at 4°C in Stl buffer (400 mM NaCl; 75 mM Hepes7.5; 5 mM MgCl$_2$). The experiment was carried out at 25°C and started by the addition of dUTP (10 µM final concentration).

## Statistical analyses

As indicated in the figure legends either a two-way ANOVA comparison with Sidak's adjustment for multiple comparisons was conducted or a one-way ANOVA, as appropriate. All analysis was done using Graphpad Prism 6 software.

## Protein expression and purification

Trimeric φ11 Dut and Stl were expressed and purified as previously described (*Maiques et al., 2016*). StlΔ$^{Cter}$ was purified following an identical protocol as for the Stl full-length protein. StlΔ$^{HTH}$ was produced from *E. coli* BL21 (DE3) (Novagen) cultures harbouring the pETNKI-StlΔ$^{HTH}$ plasmid. The culture was grown at 37°C in LB medium supplemented with 33 µg/ml kanamycin up to an OD$_{600}$ of 0.5–0.6, and then protein expression was induced with 0.1 mM isopropyl-β-D thiogalacto-pyranoside (IPTG) at 20°C for 16 hr. After induced cells were harvested by centrifugation at 4°C for 30 min at 3500 × g, the cell pellet was resuspended in buffer A (75 mM HEPES pH 7.5, 400 mM NaCl and 5 mM MgCl$_2$) supplemented with 1 mM PMSF and sonicated. A soluble fraction was obtained after centrifugation at 16 000 × g for 40 min, and it was loaded on a pre-equilibrated His Trap HP column (GE Healthcare). After washing with 10 column volumes of buffer A, the protein was eluted by adding buffer A supplemented with 500 mM imidazole. The eluted protein was digested for His-SUMO3 tag removal using SENP2 protease at a molar ratio 1:50 (protease:eluted protein) for 16 hr at 4°C with slow shaking. After digestion, the sample was loaded one more time into the pre-equilibrated His Trap HP column to remove the His-SUMO3 tag and SENP2 protease from the Stl protein. Fractions were analysed by SDS-PAGE and those fractions with purest digested Stl protein were selected, concentrated and stored at −80°C.

His-tagged dimeric φO11 Dut was overexpressed in *E. coli* BL21 (DE3) (Novagen) harbouring the pJP1938 plasmid. The cells were grown to exponential phase at 37°C in LB medium supplemented with 33 µg/ml kanamycin, and then protein expression was induced by the addition of 1 mM IPTG for 3 hr. After induction, cells were harvested by centrifugation, re-suspended in buffer A supplemented with 1 mM phenylmethanesulfonyl fluoride (PMSF) and lysed by sonication. The lysate was clarified by centrifugation and the soluble fraction was loaded on a His Trap HP column pre-equilibrated with buffer A. The column was washed with the same buffer supplemented with 10 mM imidazole and proteins were eluted with buffer A supplemented with 500 mM imidazole. The eluted proteins were concentrated and loaded onto a Superdex S200 (GE Healthcare) equilibrated with buffer B (75 mM HEPES pH 7.5, 250 mM NaCl and 5 mM MgCl$_2$) for size exclusion chromatography. The fractions were analysed by SDS-PAGE and those fractions showing purest protein were selected, concentrated and stored at −80°C.

Mass Spectrometry analyses were performed at the proteomics facility of SCSIE, University of Valencia.

## φO11 Dut crystallization and data collection

φO11 Dut protein in complex with dUPNPP protein was crystallized using the sitting drop method in the Crystallogenesis facility of IBV. φO11 (at 10 mg/ml) was incubated with 0.5 mM dUPNPP (2-Deoxyuridine-5-[(α,β)-imido]triphosphate; Jena Biosciences) and 5 mM MgCl$_2$ during 8 hr at 4°C and sitting drops were set up at 21°C. The best crystals were obtained using 0.2 M magnesium chloride, 0.1 M Tris-HCl pH8.5, 20% PEG 8000 as liquor mother. Crystals were frozen in liquid nitrogen respecting crystallization condition, increasing the cryobuffer to 35% PEG 8000 concentration for the diffraction process. Diffraction data was collected from single crystals at 100 K on ALBA (Barcelona, Spain) and DLS (Didcot, UK) synchrotrons and processed and reduced with Mosflm (*Powell et al., 2013*) and Aimless (*Evans and Murshudov, 2013*) programs from the CCP4 suite (*Winn et al., 2011*). The data-collection statistics for the best data sets used in structure determination are shown in *Supplementary file 3*.

## φO11 Dut–dUPNPP structure determination

Protein structure was solved by molecular replacement with Phaser (*McCoy et al., 2007*) and an edited PDB of the dimeric Dut from phage φDI as a model (5MYD). Based on sequence homology between φO11 and φDI Duts (70% identity), we excluded amino acids 82–140, corresponding to the divergent regions present in the phage dimeric Duts, from the starting model. This decision was

made in order to reduce the imposition of any initial structural conformation to this variable region. Iterative refinement, rebuilding and validation steps were done using programs Coot (*Emsley et al., 2010*) and Phenix (*Adams et al., 2010*). The final model includes two Dut molecules (amino acids sequence 4–160 and 3–158) forming one dimer in an asymmetric unit with one dUPNPP molecule and two Mg ions bound at each of the two active centres. The final structure has good geometry as indicated by the Ramachandran plots (any residue in the disallowed region). A summary of structure refinement statistics is shown in *Supplementary file 3*.

### Native gel mobility shift assay

Purified proteins were mixed at 40 μM 1:1 molar ratio in a buffer A (final volume 18 μl) and incubated at 4°C overnight. Samples were loaded into an 8% polyacrylamide gel and electrophoresis was performed at 4°C. Native gels were stained with coomassie brilliant blue.

### In silico protein modelling and structure comparison

The 3D homology model of 80α, φSLT, 52A and φN315 SSAPs, and SaPIbov1 Stl were constructed using I-Tasser (default mode) (*Yang et al., 2015*) and Phyre2 (intensive mode) (*Kelley et al., 2015*) servers (*Supplementary files 2A, B* and *4*). Intrinsic protein disorder was predicted by the meta-server Metadisorder (*Kozlowski and Bujnicki, 2012*).

## Acknowledgements

We would like to thank Íñigo Lasa and Wilfried Meijer for comments on the manuscript, the NKI Protein Facility for provision of LIC vectors and the IBV-CSIC Crystallogenesis Facility for protein crystallization screenings. The X-ray diffraction data reported in this work was collected in experiments performed at XALOC beamlines at ALBA Synchrotron. Preliminary and complementary X-ray diffraction experiments were performed at DLS synchrotron. We thank the staff of the beamlines used at these synchrotrons for assistance in the measurement of the crystals. This work was supported by grant BIO2013-42619-P and BIO2016-78571-P from the Ministerio de Economia y Competitividad (Spain) and grant Prometeo II/2014/029 from Valencian Government (Spain) to AM, and grants MR/M003876/1 from the Medical Research Council (UK), BB/N002873/1 from the Biotechnology and Biological Sciences Research Council (BBSRC, UK), 201531/Z/16/Z from Wellcome Trust and ERC-ADG-2014 Proposal n° 670932 Dut-signal (from EU) to JRP. EM was supported by CSIC JAE-Doc postdoctoral contract (Programa «Junta para la Ampliación de Estudios») co-funded by the European Social Fund. MMN was supported by a predoctoral fellowship from the University of Kufa and from the Ministry of Higher Education and Scientific Research (Iraq). JRC and CA were supported by FPU13/02880 and FPI BES-2014–068617 predoctoral fellowships respectively. X-ray diffraction data collection was supported by Diamond Light Source block allocation group (BAG) Proposal MX14739 and Spanish Synchrotron Radiation Facility ALBA Proposal 2015071314. The research leading to these results has received funding from the European Community's Seventh Framework Programme (FP7/2007-2013) under BioStruct-X (grant agreement N°283570).

## Additional information

### Funding

| Funder | Grant reference number | Author |
| --- | --- | --- |
| Wellcome | 201531/Z/16/Z | José R Penadés |
| Biotechnology and Biological Sciences Research Council | BB/N002873/1 | Jose R Penades |
| Medical Research Council | MR/M003876/1 | José R Penadés |
| Ministerio de Economía y Competitividad | BIO2013-42619-P | Alberto Marina |
| Ministerio de Economía y Competitividad | BIO2016-78571-P | Alberto Marina |

| ERC Advanced Grant 2014 | Proposal n° 670932 Dut-signal | Jose R Penades |

The funders had no role in study design, data collection and interpretation, or the decision to submit the work for publication.

## Author contributions

Janine Bowring, Investigation, Writing—review and editing; Maan M Neamah, Jorge Donderis, Ignacio Mir-Sanchis, Christian Alite, J Rafael Ciges-Tomas, Elisa Maiques, Iltyar Medmedov, Investigation; Alberto Marina, José R Penadés, Conceptualization, Formal analysis, Supervision, Funding acquisition, Writing—original draft, Project administration, Writing—review and editing

## Author ORCIDs

Maan M Neamah, http://orcid.org/0000-0002-7068-8416
Ignacio Mir-Sanchis, http://orcid.org/0000-0002-6536-0045
Alberto Marina, https://orcid.org/0000-0002-1334-5273
José R Penadés, http://orcid.org/0000-0002-6439-5262

## Decision letter and Author response

Decision letter https://doi.org/10.7554/eLife.26487.033
Author response https://doi.org/10.7554/eLife.26487.034

# Additional files

## Supplementary files

• Supplementary file 1. Crystallographic statistics of the φO11-dUPNPP structure.
DOI: https://doi.org/10.7554/eLife.26487.024

• Supplementary file 2. Templates and confidence values in SaPIbov1 Stl repressor models generated by I-Tasser[a] and Phyre2[b] servers.
DOI: https://doi.org/10.7554/eLife.26487.025

• Supplementary file 3. SSAPs present in *S. aureus* phages.
DOI: https://doi.org/10.7554/eLife.26487.026

• Supplementary file 4. (A) Templates and confidence values in recombinase models generated by I-Tasser server[a]. (B) Templates and confidence values in recombinase models generated by Phyre2 server[a].
DOI: https://doi.org/10.7554/eLife.26487.027

• Supplementary file 5. (A) Phage-inducible chromosomal islands analysed in this study. (B) Description and relationships between the PICI-encoded Stl repressors[a].
DOI: https://doi.org/10.7554/eLife.26487.028

• Supplementary file 6. Putative SaPI inducers are present in phages infecting species other than *S. aureus*.
DOI: https://doi.org/10.7554/eLife.26487.029

• Supplementary file 7. Inter-species PICI transfer.
DOI: https://doi.org/10.7554/eLife.26487.030

• Supplementary file 8. (A) Bacterial strains used in this study. (B) Plasmids used in this study. (C) Oligonucleotide designs used in this study.
DOI: https://doi.org/10.7554/eLife.26487.031

• Transparent reporting form
DOI: https://doi.org/10.7554/eLife.26487.032

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
