## [Decision Letter]

Thank you for submitting your article "Pirating conserved phage mechanisms promotes promiscuous staphylococcal pathogenicity island transfer" for consideration by *eLife*. Your article has been reviewed by three peer reviewers and the evaluation was overseen by Gisela Storz as the Reviewing and Senior Editor. The reviewers have opted to remain anonymous.

The reviewers have discussed the reviews with one another and the Reviewing Editor has drafted this decision to help you prepare a revised submission.

Summary:

The paper by Bowring et al. explores the mechanism of induction of *Staphylococcus aureus* Pathogenicity Islands (SaPIs). These are mobile genetic elements that carry virulence genes and, whereas they spend most of their time integrated into the staphylococcal chromosome, they can be induced during phage infection (known as the helper phage) and then propagate throughout the bacterial population. The induction and spread of SaPIs is a fascinating phenomenon. Usually helper phages produce a protein that interferes with the SaPI repressor, an event that leads to the activation of the SaPI and its excision from the chromosome previous packaging and spread. Through genetic and structural analysis, this paper reveals that a given SaPI repressor can be activated by many different phage proteins and/or domains within a phage protein. The work reveals the genetic plasticity of SaPI induction, which can be activated and spread upon multiple phage signals.

Essential revisions:

1) While all three reviewers thought your work is sound, there was less agreement on whether the findings were of sufficient general interest for *eLife*. A part of the problem is probably how the paper is written. The revised version must adequately address the following:

– The manuscript is unnecessarily difficult to read and tends to jump from one topic to another without sufficient transitions and connections. Related to this, the "big picture" is missing from the paper.

– There are many instances where the authors overstate just how "remarkable" the findings are (remarkably is used ~10 times in the manuscript, elegant/extremely sophisticated and the like are also overused), overall the language should be toned down. Some examples:

– "[…] by targeting structurally unrelated proteins as antirepressors represents a remarkable evolutionary adaptation." Why is this so remarkable? It seems more like the classical evolutionary arms race: the SaPI evolves induction through the interaction of its repressor with a phage inducer protein. This process is detrimental to phage replication and therefore the phage evolves modifications in the inducer to escape SaPI inhibition. In turn the SaPI evolves a new interaction to re-gain activation. And so on.

– "Remarkably, all of the staph phages encode one SSAP" This is not especially remarkable given that the authors indicate it is essential for replication.

– The beginning of the Discussion regarding the historical battle of Thermopylae is grandiose, and should be removed.

– The paper starts with the SaPI2 Stl repressor, suggesting that it can interact (via different domains) with structurally different recombinases. The data for this repressor-inducer is less complete and as it stands, is not sufficiently convincing. At a minimum, the authors should test truncated SaPI2 Stl repressor with the different recombinases in shift assays as they do for the Duts/SaPIbov1 Stl (even without the structural data, which they state they have not been able to get, this data is necessary to justify their conclusions). Otherwise, it seems to make more sense to reorganize the paper with the Dut story first.

2) In addition, the following controls need to be provided:

– Figure 1: The uninduced controls (T=0 as in Figure 4) should be shown for this blot (or at least an empty vector control if leaky expression). It is also not clear why in the chimera one wouldn't see the probe lighting up the bulk DNA?

– The authors use their blaZ reporter plasmid as a proxy for SaPI induction, however, the interpretation of these specific results is not justified by the experiment (i.e. "Moreover, expression of the sak4 genes prevented Stl from binding to the SaPI2 stl-str divergent region (Figure 2)". Expression from the reporter does not show that sak4 expression prevents Stl binding from that region. The authors need to perform and show the relevant experiments that would allow them to make said conclusions. While the results taken together support their conclusions, interpretation/discussion of such results individually should better represent the data. A similar statement re: Figure 2 is also found in the last paragraph of subsection “Unrelated phage-coded recombinases induce SaPI2”.

– Figure 2: as it stands there are insufficient controls shown to justify the conclusions.

– Figure 2: The "less blue" for erf is not meaningful; these BACTH assays should be quantified (especially given that the authors indicate the assays revealed "strong"' interactions). The authors should support their BACTH analysis with evidence that the chimera is being expressed, and ideally do the relevant co-purifications. (The authors mention they were unable to co-purify untagged sak4 and His tagged Stl, what about tagged sak4? What about the other proteins?)

---

## [Author Response]

Essential revisions:1) While all three reviewers thought your work is sound, there was less agreement on whether the findings were of sufficient general interest for eLife. A part of the problem is probably how the paper is written. The revised version must adequately address the following:– The manuscript is unnecessarily difficult to read and tends to jump from one topic to another without sufficient transitions and connections. Related to this, the "big picture" is missing from the paper.

Following the reviewers’ and editor’s suggestion, we have reorganised the paper. Moreover, at the end of the Introduction we have specifically highlighted what are the major questions (big picture) addressed in the manuscript.

– There are many instances where the authors overstate just how "remarkable" the findings are (remarkably is used ~10 times in the manuscript, elegant/extremely sophisticated and the like are also overused), overall the language should be toned down. Some examples:– "[…] by targeting structurally unrelated proteins as antirepressors represents a remarkable evolutionary adaptation." Why is this so remarkable? It seems more like the classical evolutionary arms race: the SaPI evolves induction through the interaction of its repressor with a phage inducer protein. This process is detrimental to phage replication and therefore the phage evolves modifications in the inducer to escape SaPI inhibition. In turn the SaPI evolves a new interaction to re-gain activation. And so on.– "Remarkably, all of the staph phages encode one SSAP" This is not especially remarkable given that the authors indicate it is essential for replication.

Corrected. The language used through the manuscript has been toned down, as suggested.

– The beginning of the Discussion regarding the historical battle of Thermopylae is grandiose, and should be removed.

While it is obvious that the most important point of a scientific article is its content, we humbly think that the way in which the stories are delivered is also a question to be considered, especially in those journals with a broader audience. Sometimes, a very nice analogy can provide a good guide for understanding a process. In this case, we really think the Thermopylae battle is a nice example to explain, to non-expert readers, how the SaPIs work. It is obvious from the reviewers’ comments that we probably did not explain this very well in the initial version of the manuscript. Hopefully we have now. But the analogy between the two stories is very clear and we really believe that this analogy helps in highlighting the strategy that the SaPIs use to be widespread in nature. For this reason, we would like to maintain this comparison in the text.

– The paper starts with the SaPI2 Stl repressor, suggesting that it can interact (via different domains) with structurally different recombinases. The data for this repressor-inducer is less complete and as it stands, is not sufficiently convincing. At a minimum, the authors should test truncated SaPI2 Stl repressor with the different recombinases in shift assays as they do for the Duts/SaPIbov1 Stl (even without the structural data, which they state they have not been able to get, this data is necessary to justify their conclusions). Otherwise, it seems to make more sense to reorganize the paper with the Dut story first.

As suggested by the reviewers, we initially tried many different strategies to map the interacting domains between the different recombinases and the SaPI2 Stl repressor. Unfortunately, as indicated in the text, apart from Sak the rest of the recombinases are insoluble, as is the Stl when purified alone. For this reason we have addressed the question using the Duts.

Following the reviewers’ suggestion, we have reorganised the manuscript, starting with the Duts.

2) In addition, the following controls need to be provided:– Figure 1: The uninduced controls (T=0 as in Figure 4) should be shown for this blot (or at least an empty vector control if leaky expression). It is also not clear why in the chimera one wouldn't see the probe lighting up the bulk DNA?

The figure (now Figure 5) has been modified and now it includes an empty plasmid (and the chimera protein) as controls. In addition to that, the different recombinases have been 3xflag-tagged, so a western-blot could be performed showing all the proteins are expressed.

In addition to chromosomal and phage DNA, the bulk DNA contains replicating (concatemeric) SaPI DNA. Since SaPI2 induction by the recombinases is quite effective, and since the exposure of the Southern blot is short, this explains why the negative controls do not show any signal in the bulk DNA. The composition of the bulk DNA has now been specified in the figure legend.

– The authors use their blaZ reporter plasmid as a proxy for SaPI induction, however, the interpretation of these specific results is not justified by the experiment (i.e. "Moreover, expression of the sak4 genes prevented Stl from binding to the SaPI2 stl-str divergent region (Figure 2)". Expression from the reporter does not show that sak4 expression prevents Stl binding from that region. The authors need to perform and show the relevant experiments that would allow them to make said conclusions. While the results taken together support their conclusions, interpretation/discussion of such results individually should better represent the data. A similar statement re: Figure 2 is also found in the last paragraph of subsection “Unrelated phage-coded recombinases induce SaPI2”.

The text has been modified accordingly. We have now specified that overexpression of the SaPI inducers activates expression of Stl repressed genes.

– Figure 2: as it stands there are insufficient controls shown to justify the conclusions.

Different additional controls have been used to demonstrate that the interaction is specific. These include the use of an untagged ORF16 and the use of a tagged chimeric protein (overexpressed with the SaPI2 Stl repressor). All this information is included in the text.

– Figure 2: The "less blue" for erf is not meaningful; these BACTH assays should be quantified (especially given that the authors indicate the assays revealed "strong"' interactions). The authors should support their BACTH analysis with evidence that the chimera is being expressed, and ideally do the relevant co-purifications. (The authors mention they were unable to co-purify untagged sak4 and His tagged Stl, what about tagged sak4? What about the other proteins?)

As previously mentioned, all the recombinases (except Sak) were insoluble when purified. That is the reason we performed the BACTH assays. Incidentally, we have now quantified the BACTH assays (see new Figure 6). The result confirms the interaction between the recombinases (except the chimera) and the SaPI2 Stl repressor.

We have now shown that all the recombinases (including the chimera) are expressed. As is also mentioned in the text, there is evidence that all these proteins are expressed in vivo, since the recombinases are essential for phage replication.